# The Nonlinear Eigenvalue Problem of Electromagnetic Wave Propagation in a Dielectric Layer Covered with Graphene

Yury Smirnov *[ID] and Stanislav Tikhov [ID]

Mathematics and Supercomputing, Penza State University, 440026 Penza, Russia; tik.stanislav2015@yandex.ru
* Correspondence: smirnovyug@mail.ru

**Abstract:** The paper focuses on the problem of a monochromatic terahertz TE-polarized wave propagation in a plane dielectric layer filled with a homogeneous isotropic medium; one of the boundaries of the waveguide is covered with a layer of graphene. In fact, the paper aims to find the eigenwaves of the described waveguiding structure. On the one hand, in the study, energy losses both in the dielectric layer and in the graphene layer are neglected; the latter assumption is reasonable in the terahertz range of electromagnetic radiation (on which the paper focuses), where graphene has a strong plasmonic response and much less loss. On the other hand, this study takes into account the significant third-order nonlinearity resulting from the interaction of the electromagnetic wave with the charge carriers in the graphene layer. The paper aims to study the guiding properties of the above structure using primarily an analytical approach. The wave propagation problem is reduced to an eigenvalue problem, where one of the boundary conditions is nonlinear with respect to the sought-for function. The main result of the paper is a dispersion equation allowing for a waveguide of a given thickness to determine a set of its propagation constants and, consequently, a set of its eigenwaves. It is worth noting that the dispersion equation being written in an explicit form can be used to obtain deep qualitative results related to the solvability of the problem and the properties of its solutions. For example, in the paper, the existence of several propagation constants (and, consequently, the eigenwaves) of the studied waveguiding structure is proved under some conditions. Besides studying the problem analytically, the paper presents some numerical results as well. In particular, the presented figures demonstrate how the nonlinearity in graphene affects the propagation constants and eigenwaves, providing the dispersion curves and eigenwaves for nonlinear graphene as well as for the linear one.

**Keywords:** Maxwell's equations; electromagnetic waves; plane dielectric layer; graphene; nonlinear surface electric conductivity; eigenvalue problem; boundary value problem





## 1. Introduction

Today, much attention is being paid to 2D materials and 2D electronic components; among the two-dimensional materials, graphene, discovered in 2004 by Geim and Novoselov [1], occupies a special place. Graphene is a hexagonal crystal lattice of carbon atoms; due to its special structure, graphene has a number of unique physical properties. In particular, graphene has properties that make it very promising for various applications in photonics and optoelectronics: combining graphene with other materials, one can create a variety of waveguiding devices with better characteristics compared to waveguides made of «traditional» materials. In [2], there is an overview of waveguides of different configurations based on the use of graphene; such waveguides can serve as components for various optical devices such as photodetectors, modulators, polarizers, etc., significantly increasing their efficiency. In [3–5], waveguiding structures with graphene are shown to be efficient in many fields, such as biosensors, photoelectric detection, and so on; in [6], the authors considered graphene oxide (GO) as a potential absorbent for environmental cleanup and suggested an approach, which enables making full use of GO for this purpose.

The above applications use only the «linear» optical and electrical properties of graphene. In papers [7,8], it was theoretically predicted, and in [9], it was experimentally confirmed that graphene has an extremely strong third-order nonlinearity compared to the commonly used dielectrics and metals; this nonlinearity results from the interaction of the charge carriers in graphene with strong electromagnetic radiation. The discovery of the strong nonlinear response of graphene has led to significant efforts to realize a new generation of practical nonlinear photonic guided wave devices for all optical applications [10].

This paper focuses on the propagation of a monochromatic terahertz TE-polarized electromagnetic wave in a plane dielectric waveguide, one of the surfaces of which is covered with a layer of graphene. In fact, in this paper, we consider the problem of finding the eigenmodes of the waveguide, i.e., such electromagnetic waves that can propagate in the waveguide without obtaining energy from any external source. In eigenwave problems, energy losses in the waveguiding structure are often neglected, assuming that the energy losses are small, and the propagation length of the found eigenwaves is sufficient in practical applications. Such an approach is also applied in this paper: we neglect the energy losses in the dielectric as well as in the graphene layer. We stress that although in the general case graphene absorbs some of the incident electromagnetic wave, in the terahertz range, the absorption is small compared to the strong plasmonic response of graphene. We understand that neglecting the absorption properties of graphene might seem to be a serious drawback of this study. However, we stress that this requirement is unavoidable in our statement of the problem. Section 4 deals with this issue in more detail.

For applications in photonics and optoelectronics, a convenient parameter characterizing graphene is its surface electrical conductivity $\sigma_g$; it is very important that the conductivity of graphene can be controlled by an electric field or chemical doping without changing the size and structure of the sample. In accordance with [7–9,11], the conductivity $\sigma_g$ of graphene depends on the strength of the electric field interacting with graphene. Due to the centrally symmetric structure of graphene, this dependence has the form $\sigma_g = \sigma^{(1)} + \sigma^{(3)}|\mathbf{E}|^2$, where $\sigma^{(1)}$ and $\sigma^{(3)}$ are quantities depending on the wave frequency. As the electric conductivity differs from zero, there arises a surface current. Since these phenomena are localized on the surface, it is essential to take them into account in the boundary conditions of the problem. Taking into account the nonlinear effects arising in the graphene ($\sigma_3 \neq 0$), one obtains the boundary conditions, which are nonlinear with respect to the sought-for function, which, in turn, brings additional complexity to the problem.

The guiding properties of graphene structures play an important role for practical applications. The ability to propagate TM- and TE-polarized waves localized on a graphene monolayer with the dispersion in the terahertz range of electromagnetic radiation is shown in [12–14]. The ability of the structure formed by two graphene layers and a separating thin dielectric layer to hold the localized plasmon modes has been studied in many papers [15–17]. In [18], the authors investigated the features of guided TE wave modes in a plane structure consisting of a set of alternating layers of dielectric and graphene. We stress that the novelty of this paper is that the guiding properties of the considered waveguiding structure (plane dielectric waveguide covered with a graphene layer) are studied taking into account the abovementioned third-order nonlinearity of graphene. The obtained results show that this nonlinearity affects the wave propagation process significantly (for strong electromagnetic radiation) and for this reason cannot be neglected.

## 2. Materials and Methods

The main physical model we deal with in this study is the system of Maxwell's equations describing the dynamics of an electromagnetic field. Since we assume that everywhere there is no bulk current (the medium in the waveguide is not conducting) and an uncompensated electric charge, and the electromagnetic field harmonically depends on time, then the system of Maxwell's equations can be written in the harmonic mode, see Formula (3). One of the boundaries of the considered waveguiding structure is covered with graphene, which implies a surface current when an electromagnetic wave propagates.

Since this phenomena is localized on the surface, it is essential to take it into account in the boundary conditions of the problem. Graphene is characterized by its surface electric conductivity $\sigma_g$. In this study, we take into account the nonlinear optical properties of graphene supposing that $\sigma_g$ depends on the electric field strength, see Formula (5).

The main goal of the paper is to study the guiding properties of the considered waveguiding structure analytically. To do this, the problem on electromagnetic wave propagation is reduced to an eigenvalue problem, where the eigenvalues correspond to the propagation constants. Solving the main differential equation and taking into account the boundary conditions, we obtain the dispersion equation of the problem written in an explicit form. Studying this equation analytically, we obtain results related to the solvability of the problem and the properties of its solutions.

For solving the eigenvalue problem $\mathcal{P}$, see Section 3.2, numerically, we use mainly the so-called «shooting method». The main schema of the method is following. We fix some segment on $\gamma$, say $\gamma \in [\gamma', \gamma'']$, and generate a grid with nodes $\gamma' \leqslant \gamma_1 < \gamma_2 < \ldots < \gamma_n \leqslant \gamma''$. For each $\gamma = \gamma_i$, we solve the Cauchy problem for equation $u''(x) = -(\varepsilon_2 - \gamma^2)u(x)$ with the initial conditions $u(0) = A$, $u'(0) - \sqrt{\gamma^2 - \varepsilon_1} \cdot u(0) = 0$ and evaluate its solution $u \equiv u(x; \gamma_i)$ and the first derivative $u'(x; \gamma_i)$ of this solution at point $x = h$. Then, going through all $\gamma_i$, we look for the condition $F(\gamma_i) \cdot F(\gamma_{i+1}) < 0$, where

$$F(\gamma) = u'(h; \gamma) + \sqrt{\gamma^2 - \varepsilon_3} \cdot u(h; \gamma) + 120\pi i\Big(\sigma^{(1)} + \sigma^{(3)} u^2(h; \gamma)\Big)u(h; \gamma),$$

is true; if it is true, then segment $\gamma \in [\gamma_i, \gamma_{i+1}]$ definitely contains a solution to problem $\mathcal{P}$. The numerical results are presented in Section 3.5. All the numerical methods are implemented with the package «Maple».

## 3. Results

### 3.1. Electrodynamic Statement of the Problem

A monochromatic TE-polarized electromagnetic wave

$$(\mathbf{E}, \mathbf{H})e^{-i\omega t}, \tag{1}$$

where $\omega$ is a circular frequency,

$$\mathbf{E} = \big(0, \mathrm{E}_y(x), 0\big)e^{i\gamma z}, \quad \mathbf{H} = \big(\mathrm{H}_x(x), 0, \mathrm{H}_z(x)\big)e^{i\gamma z} \tag{2}$$

are complex amplitudes, and $\gamma$ is (unknown) real spectral parameter, propagates in a plane dielectric layer

$$\Sigma = \{(x, z) \in \mathbb{R}^2 : 0 \leqslant x \leqslant h\}$$

sandwiched between two half-spaces $x < 0$ and $x > h$. The boundary $x = 0$ is open, and at the boundary $x = h$, there is a graphene layer, which causes a surface current at this boundary.

Waveguide $\Sigma$ is filled with a homogeneous isotropic medium characterized by a constant permittivity $\varepsilon_2$; half-spaces $x < 0$ and $x > h$ are filled with homogeneous isotropic mediums characterized by constant permittivities $\varepsilon_1$ and $\varepsilon_3$, respectively, such that $1 \leqslant \varepsilon_1 \leqslant \varepsilon_3 \leqslant \varepsilon_2$. Everywhere, permeability $\mu = \mu_0$, where $\mu_0$ is the magnetic constant. The geometry of the problem is presented in Figure 1.

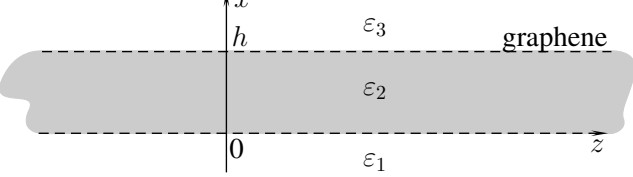

**Figure 1.** Geometry of the problem.

Field (1) satisfies Maxwell's equations

$$\text{rot}\,\mathbf{H} = -i\omega\varepsilon_0\varepsilon\mathbf{E}, \quad \text{rot}\,\mathbf{E} = i\omega\mu\mathbf{H}, \tag{3}$$

where $\varepsilon_0$ is the dielectric constant, and

$$\varepsilon = \begin{cases} \varepsilon_1, & x < 0, \\ \varepsilon_2, & 0 < x < h, \\ \varepsilon_3, & x > h. \end{cases}$$

Since the boundaries of the waveguide are open, and the electromagnetic field penetrates into half-spaces $x < 0$ and $x > h$, it is essential to impose on (1) the condition of decaying at $x \to \pm\infty$. The tangential component $\mathrm{E}_y$ of the electric field is continuous at both boundaries, due to the absence of a surface charge there. The tangential component of the magnetic field is also continuous at the boundary $x = 0$, due to the absence of a surface current there; however, it undergoes a jump at the boundary $x = h$, due to the surface current of charge carried in graphene (the current is induced by the electromagnetic wave), and the jump is equal to surface current density $\mathbf{j}$. Thus, at $x = h$, the discussed component satisfies the following condition

$$[\mathrm{n}, \mathbf{H}^+ - \mathbf{H}^-] = \mathbf{j} = \sigma_g\mathbf{E}, \tag{4}$$

where n is a unit vector of the normal directed along the $x$ axis, $\mathbf{H}^+$ and $\mathbf{H}^-$ are the values of magnetic field above and below the surface $x = h$, respectively, $[*, *]$ is the vector product, and $\sigma_g$ is the surface conductivity of graphene.

As was previously noted, the electric conductivity $\sigma_g$ of graphene depends on the electric field coupling to the charge carriers in graphene; due to the central symmetric structure of graphene, this dependence has the form

$$\sigma_g = \sigma^{(1)} + \sigma^{(3)}|\mathbf{E}|^2, \tag{5}$$

where $\sigma^{(1)}$ and $\sigma^{(3)}$ are generally some complex numbers depending on frequency $\omega$ [7–9,19,20].

The linear part $\sigma^{(1)}$ of graphene's electric conductivity is determined by formulas [21,22]. We assume that $\Re\sigma^{(1)} = 0$ and $\Im\sigma^{(1)} > 0$, i.e.,

$$\sigma^{(1)} = i\sigma', \quad \sigma' > 0. \tag{6}$$

Such a restriction on $\sigma_1$ is fair in THz range, where graphene has a strong plasmonic response and much less loss [23,24]. For determining $\sigma^{(3)}$ in (5), there exist several formulas [7,19,20]; for example, one can apply the formula in [7]; in accordance with this formula, quantity $\sigma^{(3)}$ is purely imaginary and $\Im\sigma^{(3)} < 0$, i.e.,

$$\sigma^{(3)} = -i\sigma'', \quad \sigma'' > 0. \tag{7}$$

The main problem — we call it problem $\mathcal{P}'$ — is to find such values of parameter $\gamma = \gamma'$, where there exists field (1) satisfying Maxwell's equations (3) and all the above conditions and exponentially decaying at $x \to \pm\infty$.

Substituting (2) into Maxwell's equations (3), one obtains

$$\begin{cases} i\gamma\mathrm{H}_x(x) - \mathrm{H}_z'(x) = -i\omega\varepsilon_0\varepsilon\mathrm{E}_y(x), \\ -i\gamma\mathrm{E}_y(x) = i\omega\mu\mathrm{H}_x(x), \\ \mathrm{E}_y'(x) = i\omega\mu\mathrm{H}_z(x). \end{cases} \tag{8}$$

The third equation in the obtained system provides the relationship between the tangential components of the electric and magnetic fields expressed by the formula

$$H_z(x) = -\frac{i}{\omega\mu}E'_y(x).\tag{9}$$

Expressing $H_x$ and $H'_z$ from the second and the third equations of system (8), respectively, and substituting them into the first equation, one obtains

$$\gamma^2 E_y(x) - E''_y(x) = \omega^2\mu\varepsilon_0\varepsilon E_y(x).$$

Passing to dimensionless variables in the obtained equation by virtue of the formulas

$$\widetilde{x} = k_0 x, \quad \widetilde{h} = k_0 h, \quad \widetilde{\gamma} = k_0^{-1}\gamma,$$

where $k_0^2 = \omega^2\mu\varepsilon_0$, and, omitting the tilde, we obtain the equation

$$E''_y(x) = -(\varepsilon - \gamma^2)E_y(x).\tag{10}$$

Solving (10) in half-spaces $x < 0$, $x > h$ and using the condition at infinity, one finds

$$E_y(x) = \begin{cases} C_1 e^{x\sqrt{\gamma^2-\varepsilon_1}}, & x < 0, \\ C_2 e^{-(x-h)\sqrt{\gamma^2-\varepsilon_3}}, & x > h, \end{cases}\tag{11}$$

where $C_1$ and $C_2$ are constants. Note that $E_y$ must be real-valued; from here, taking into account inequality $\varepsilon_3 \geqslant \varepsilon_1$, it follows that $\gamma$ must satisfy

$$\gamma^2 > \varepsilon_3.\tag{12}$$

From the above conditions imposed on field (1) and Formula (9), expressing the relationship between the tangential components of the magnetic and electric fields, it follows that $E_y$ must satisfy the conditions

$$\begin{aligned} E_y|_{0-0} - E_y|_{0+0} &= 0, \\ E'_y|_{0-0} - E'_y|_{0+0} &= 0 \end{aligned}\tag{13}$$

and

$$\begin{aligned} E_y|_{h-0} - E_y|_{h+0} &= 0, \\ E'_y|_{h-0} - E'_y|_{h+0} &= -i \cdot 120\pi\sigma_g E_y|_{x=h}, \end{aligned}\tag{14}$$

where

$$\sigma_g = \sigma^{(1)} + \sigma^{(3)} E_y^2\Big|_{x=h},\tag{15}$$

which results from (5), taking into account the form of field (2) and the realness of parameter $\gamma$.

Then, one can reformulate the problem $\mathcal{P}'$ in the following way. Problem $\mathcal{P}'$ is to find such values $\gamma = \gamma'$ satisfying inequality (12), where there exists the solution $E_y \equiv E_y(x; \gamma')$ to Equation (10), satisfying the boundary conditions (13) and (14).

### 3.2. An Eigenvalue Problem

Introducing the notation $u(x) := E_y(x)$, Equation (10) in the layer can be written in the form

$$u''(x) = -(\varepsilon_2 - \gamma^2)u(x).\tag{16}$$

Using conditions (13) and (14), the solutions in half-spaces (11), and Formula (15), one can write the boundary conditions for function $u(x)$ as

$$u'(0) - \sqrt{\gamma^2 - \varepsilon_1} \cdot u(0) = 0,\tag{17}$$

$$u'(h) + \sqrt{\gamma^2 - \varepsilon_3} \cdot u(h) = -120\pi i(\sigma^{(1)} + \sigma^{(3)} u^2(h))u(h).\tag{18}$$

Besides this, we need one more condition in order to obtain a discrete set of solutions to the problem. It is in accordance with the physical process of the electromagnetic wave propagation in waveguiding structures. We use the following form of the additional condition

$$u(0) = A, \tag{19}$$

where $A \neq 0$ is a constant.

It is easy to determine that when a couple $(\gamma, u)$ satisfies Equation (16), $(-\gamma, u)$ satisfies it as well. For this reason, we consider positive $\gamma$.

So, problem $\mathcal{P}'$ is equivalent to the boundary value problem; we call it problem $\mathcal{P}$, which is to find the positive $\gamma = \widehat{\gamma}$ satisfying inequality (12), such that there exists a twice continuously differentiable function $u \equiv u(x; \widehat{\gamma})$ that is a solution to Equation (16) satisfying conditions (17)–(19). A number $\widehat{\gamma}$ is called an *eigenvalue* of problem $\mathcal{P}$, and function $u(x; \widehat{\gamma})$ is called an *eigenfunction* of problem $\mathcal{P}$.

Note that condition (18) is nonlinear with respect to the unknown function $u(x)$; so, in fact problem $\mathcal{P}$ is nonlinear. If one sets $\sigma^{(3)} = 0$ in (18), then problem $\mathcal{P}$ degenerates into a linear problem, which we call problem $\mathcal{P}_0$.

Although the statement of problem $\mathcal{P}$ is given above, and problem $\mathcal{P}_0$ is its special case, we present the statement of problem $\mathcal{P}_0$ here as well. So, problem $\mathcal{P}_0$ is to find the positive $\gamma = \widetilde{\gamma}$ satisfying inequality (12), such that there exists a function $v \equiv v(x; \gamma)$, which is a solution to the equation

$$v''(x) = -(\varepsilon_2 - \gamma^2)v(x)$$

satisfying the following boundary conditions

$$v'(0) - \sqrt{\gamma^2 - \varepsilon_1} \cdot v(0) = 0,$$
$$v'(h) + \sqrt{\gamma^2 - \varepsilon_3} \cdot v(h) = -i \cdot 120\pi\sigma^{(1)}v(h);$$

the number $\gamma = \widetilde{\gamma}$ is called an eigenvalue of problem $\mathcal{P}_0$, and function $v(x; \widetilde{\gamma})$ is called an eigenfunction of problem $\mathcal{P}_0$.

We stress that problem $\mathcal{P}_0$ does not require (19).

Problem $\mathcal{P}_0$ and problem $\mathcal{P}$ serve as mathematical models of the monochromatic TE-wave propagation in a plane waveguide having a graphene layer on one of its boundaries, but in the case of problem $\mathcal{P}_0$ graphene is characterized by $\sigma_g = \sigma_1$, i.e., the nonlinear response of graphene is neglected. Note that such a model allows one to obtain results close to the experimental data only at intensities of electromagnetic radiation.

### 3.3. Dispersion Equation of Problem $\mathcal{P}$

The solution to Equation (16) has the form

$$u = C_3 \cos \sqrt{\varepsilon_2 - \gamma^2}x + C_4 \sin \sqrt{\varepsilon_2 - \gamma^2}x,$$

where $C_3$ and $C_4$ are constants. Using condition (17), one can express $C_4$ by $C_3$ and then, using condition (19), determine $C_3$; doing this, one obtains

$$u = \frac{1}{\sqrt{\varepsilon_2 - \gamma^2}}A\left(\sqrt{\gamma^2 - \varepsilon_1} \sin \sqrt{\varepsilon_2 - \gamma^2}x + \sqrt{\varepsilon_2 - \gamma^2} \cos \sqrt{\varepsilon_2 - \gamma^2}x\right).$$

Using condition (18), one obtains the following equation

$$\left(\sqrt{\gamma^2 - \varepsilon_1}\sqrt{\gamma^2 - \varepsilon_3} + 120\pi i \sigma^{(1)}\sqrt{\gamma^2 - \varepsilon_1} - (\varepsilon_2 - \gamma^2)\right)\sin h\sqrt{\varepsilon_2 - \gamma^2} +$$

$$+ \sqrt{\varepsilon_2 - \gamma^2}\left(\sqrt{\gamma^2 - \varepsilon_1} + \sqrt{\gamma^2 - \varepsilon_3} + 120\pi i \sigma^{(1)}\right)\cos h\sqrt{\varepsilon_2 - \gamma^2} =$$

$$= -120\pi i \frac{A^2}{\varepsilon_2 - \gamma^2}\left(\sqrt{\gamma^2 - \varepsilon_1}\sin h\sqrt{\varepsilon_2 - \gamma^2} + \sqrt{\varepsilon_2 - \gamma^2}\cos h\sqrt{\varepsilon_2 - \gamma^2}\right)^3. \quad (20)$$

Let us introduce notation $\lambda^2 = \varepsilon_2 - \gamma^2$, $\epsilon_1^2 = \varepsilon_2 - \varepsilon_1$, $\epsilon_3^2 = \varepsilon_2 - \varepsilon_3$, $\bar{\sigma}_1 = i \cdot 120\pi\sigma^{(1)}$, $\bar{\sigma}_3 = i \cdot 120\pi\sigma^{(3)}$; taking into account Formulas (6) and (7), it is clear that $\bar{\sigma}_1$ and $\bar{\sigma}_3$ are real, $\bar{\sigma}_1 < 0$, and $\bar{\sigma}_3 > 0$. Using this notation, Equation (20) can be written in the form

$$\left(\sqrt{\epsilon_1^2 - \lambda^2}\sqrt{\epsilon_3^2 - \lambda^2} - |\bar{\sigma}_1|\sqrt{\epsilon_1^2 - \lambda^2} - \lambda^2\right)\sin \lambda h +$$

$$+ \left(\sqrt{\epsilon_3^2 - \lambda^2} + \sqrt{\epsilon_1^2 - \lambda^2} - |\bar{\sigma}_1|\right)\lambda\cos \lambda h =$$

$$= -|\bar{\sigma}_3|\lambda^{-2}A^2\left(\sqrt{\epsilon_1^2 - \lambda^2}\sin \lambda h + \lambda\cos \lambda h\right)^3. \quad (21)$$

Expression (21) can be considered as the characteristic equation of problem $\mathcal{P}$. This means that any solution $\lambda = \widehat{\lambda}$ to Equation (21) corresponds to an eigenvalue $\widehat{\gamma} = \sqrt{\varepsilon_2 - \widehat{\lambda}^2}$ of problem $\mathcal{P}$, and any eigenvalue $\widehat{\gamma}$ of problem $\mathcal{P}$ corresponds to the solution $\widehat{\lambda} = \sqrt{\varepsilon_2 - \widehat{\gamma}^2}$ of Equation (21).

From a physical point of view, relation (21) is the so-called dispersion equation, as it provides the relationship between the thickness of the waveguide and its propagation constants.

Setting $\bar{\sigma}_3 = 0$ in (21), one obtains the dispersion equation for problem $\mathcal{P}_0$ in the form

$$\text{tg}\,\lambda h = \frac{\lambda\left(\sqrt{\epsilon_3^2 - \lambda^2} + \sqrt{\epsilon_1^2 - \lambda^2} - |\bar{\sigma}_1|\right)}{\lambda^2 - \sqrt{\epsilon_1^2 - \lambda^2}\cdot\sqrt{\epsilon_3^2 - \lambda^2} + |\bar{\sigma}_1|\sqrt{\epsilon_1^2 - \lambda^2}}. \quad (22)$$

Further, setting $\bar{\sigma}_1 = 0$ in the previous formula, one obtains the classical dispersion equation of the form

$$\text{tg}\,\lambda h = \frac{\lambda\left(\sqrt{\epsilon_3^2 - \lambda^2} + \sqrt{\epsilon_1^2 - \lambda^2}\right)}{\lambda^2 - \sqrt{\epsilon_1^2 - \lambda^2}\cdot\sqrt{\epsilon_3^2 - \lambda^2}} \quad (23)$$

for the problem of the electromagnetic TE-wave propagation in a plane dielectric layer sandwiched between two half-spaces; the above equation is given in [25] and in [26] (only for the case $\epsilon_1 = \epsilon_3$).

### 3.4. Solvability of Problem $\mathcal{P}$

Problem $\mathcal{P}$ might have eigenvalues in the interval $\gamma \in (\sqrt{\varepsilon_3}, \sqrt{\varepsilon_2})$, as well as in the unbounded domain $\gamma > \sqrt{\varepsilon_2}$ (unless the contrary is proved). The former kind of eigenvalues corresponds to solutions of Equation (21) belonging to the interval $\lambda \in \Lambda$, where $\Lambda = (0, \epsilon_3)$; the latter one corresponds to solutions of Equation (21) having the form $\lambda = i\lambda'$ with $\lambda' > 0$.

**Statement 1.** *Equation (21) does not have solutions of the form $\lambda = i\lambda'$, $\lambda' > 0$ if $\lambda' > \lambda'_*$, where*

$$\lambda'_* = \max\left\{\frac{\sqrt{|\bar{\sigma}_1|^2 + |\bar{\sigma}_1|\sqrt{|\bar{\sigma}_1|^2 + 16\epsilon_1^2}}}{2\sqrt{2}}, \sqrt{\frac{|\bar{\sigma}_1|}{2}}\right\}. \quad (24)$$

**Proof.** So, let $\lambda = i\lambda'$, where $\lambda' > 0$. Substituting $\lambda = i\lambda'$ into (21), one obtains the equation, with respect to $\lambda'$, of the form

$$\left(\sqrt{\epsilon_1^2 + \lambda'^2}\sqrt{\epsilon_3^2 + \lambda'^2} - |\bar{\sigma}_1|\sqrt{\epsilon_1^2 + \lambda'^2} + \lambda'^2\right)\operatorname{sh}\lambda'h +$$

$$+ \left(\sqrt{\epsilon_3^2 + \lambda'^2} + \sqrt{\epsilon_1^2 + \lambda'^2} - |\bar{\sigma}_1|\right)\lambda'\operatorname{ch}\lambda'h =$$

$$= -|\bar{\sigma}_3|\lambda'^{-2}A^2\left(\sqrt{\epsilon_1^2 + \lambda'^2}\operatorname{sh}\lambda'h + \lambda'\operatorname{ch}\lambda'h\right)^3. \quad (25)$$

It is clear that the right-hand side of (25) is negative for all $\lambda' > 0$, whereas the left-hand side is positive starting with some $\lambda' = \lambda'_*$. This means that Equation (25) does not have solutions in the domain $\lambda' > \lambda'_*$.

Let us obtain the estimate (24) for $\lambda'_*$. It is clear that the left-hand side is positive as soon as both terms in brackets are positive. Taking into account the inequality

$$\sqrt{\epsilon_1^2 + \lambda'^2}\sqrt{\epsilon_3^2 + \lambda'^2} - |\bar{\sigma}_1|\sqrt{\epsilon_1^2 + \lambda'^2} + \lambda'^2 \geqslant 2\lambda'^2 - |\bar{\sigma}_1|\sqrt{\epsilon_1^2 + \lambda'^2}$$

and the inequality $2\lambda'^2 - |\bar{\sigma}_1|\sqrt{\epsilon_1^2 + \lambda'^2} \geqslant 0$, which holds for all $\lambda' \geqslant \lambda'_1$, where

$$\lambda'_1 = \frac{\sqrt{|\bar{\sigma}_1|^2 + |\bar{\sigma}_1|\sqrt{|\bar{\sigma}_1|^2 + 16\epsilon_1^2}}}{2\sqrt{2}},$$

one finds that the expression in the first bracket is positive for all $\lambda' > \lambda'_1$. Further, the inequality

$$\sqrt{\epsilon_3^2 + \lambda'^2} + \sqrt{\epsilon_1^2 + \lambda'^2} - |\bar{\sigma}_1| \geqslant 2\lambda' - |\bar{\sigma}_1|$$

together with $2\lambda' - |\bar{\sigma}_1| \geqslant 0$, which holds for all $\lambda' \geqslant \lambda'_2$, where $\lambda'_2 = \sqrt{\frac{|\bar{\sigma}_1|}{2}}$, imply that the expression in the second bracket is positive for all $\lambda' > \lambda'_2$. Combining the obtained results, one obtains Formula (24). $\square$

Now, let us pass to the case $\lambda \in \Lambda$. For the further analysis, it is convenient to rewrite Equation (21) in the following way

$$\operatorname{tg}\lambda h = -\frac{\lambda \cdot \theta_1(\lambda)}{\theta_2(\lambda)}, \quad (26)$$

where

$$\theta_1(\lambda) = \sqrt{\epsilon_1^2 - \lambda^2} + \sqrt{\epsilon_3^2 - \lambda^2} - |\bar{\sigma}_1| + |\bar{\sigma}_3|A^2\left(1 + \frac{3\epsilon_1^2 - 4\lambda^2}{\lambda^2}\sin^2\lambda h\right),$$

$$\theta_2(\lambda) = \sqrt{\epsilon_1^2 - \lambda^2}\left[\sqrt{\epsilon_3^2 - \lambda^2} - |\bar{\sigma}_1| + |\bar{\sigma}_3|A^2\left(3\cos^2\lambda h + \frac{\epsilon_1^2 - \lambda^2}{\lambda^2}\sin^2\lambda h\right)\right] - \lambda^2.$$

Since functions $\theta_1$ and $\theta_2$ in the right-hand side of (26) do not have accumulation points of zeros, then (26) can have only a finite number of solutions $\lambda \in \Lambda$.

The following result provides a sufficient condition for the existence of at least one solution to Equation (26).

**Statement 2.** *If inequalities*

$$|\sigma_3|A^2\min\left\{3; \epsilon_1^2\epsilon_3^{-2}\right\} > |\sigma_1| \quad (27)$$

*and*

$$h > \frac{3\pi\sqrt{\epsilon_1^2 + \epsilon_3^2}}{2\epsilon_1 \epsilon_3} \tag{28}$$

*are fulfilled, then Equation (26) has at least one solution $\widehat{\lambda} \in (0, \lambda_*)$, where*

$$\lambda_* = \frac{\epsilon_1 \epsilon_3}{\sqrt{\epsilon_1^2 + \epsilon_3^2}} (< \epsilon_3). \tag{29}$$

**Proof.** Let us consider the right-hand side of (26). It is clear that function $\theta_1(\lambda)$ is continuous for all $\lambda \in \Lambda$. Function $\theta_2(\lambda)$ is continuous for all $\lambda \in \Lambda$, but generally, it can change the sign. Nevertheless, one can check that under condition (27), function $\theta_2$ preserves the sign and, to be more precise, is positive for $\lambda < \lambda_*$, where $\lambda_*$ is defined in (29). Indeed, taking into account the chain of simple inequalities

$$3\cos^2 \lambda h + \frac{\epsilon_1^2 - \lambda^2}{\lambda^2} \sin^2 \lambda h \geqslant 3\cos^2 \lambda h + \frac{\epsilon_1^2 - \lambda_*^2}{\lambda_*^2} \sin^2 \lambda h \geqslant \min\left\{3; \epsilon_1^2 \epsilon_3^{-2}\right\}$$

and condition (27), one can see that

$$|\bar{\sigma}_3| A^2 \left(3\cos^2 \lambda h + \frac{\epsilon_1^2 - \lambda^2}{\lambda^2} \sin^2 \lambda h\right) > |\bar{\sigma}_1|.$$

Combining this result with the inequality

$$\sqrt{\epsilon_1^2 - \lambda^2}\sqrt{\epsilon_3^2 - \lambda^2} \geqslant \lambda^2$$

taking place for $\lambda < \lambda_*$, one comes to conclusion that $\theta_2(\lambda) > 0$ for such $\lambda$. Thus, the right-hand side in (26) is continuous for $\lambda < \lambda^*$.

Condition (28) implies that, firstly, $\lambda h$ takes all values from zero to $3\pi/2$, and secondly, inequality $\lambda^* h > 3\pi/2$ is valid. Equation (26) is defined for $\lambda h \in (0, 3\pi/2)$, and the right-hand side of (26) is continuous for such $\lambda h$. This means that a graph of the function in the right-hand side of (26) has at least one intersection with tg $\lambda h$ in the left-hand side of (26), for $\lambda \in (h^{-1}\pi/2, h^{-1}3\pi/2)$. $\quad\square$

The sufficient condition for the existence of one solution to Equation (26) given in Statement 2 can easily be generalized to the case of the existence of $n > 1$ solutions. Indeed, the following result takes place.

**Statement 3.** *If inequalities (27) and*

$$h > \frac{(1 + 2n)\pi\sqrt{\epsilon_1^2 + \epsilon_3^2}}{2\epsilon_1 \epsilon_3} \tag{30}$$

*are fulfilled, then Equation (26) has at least n solutions $\widehat{\lambda}_k \in (0, \lambda_*)$, $k = \overline{1, n}$, where $\lambda_*$ is defined in (29).*

**Proof.** The proof of this statement repeats the proof of Statement 2. We have $\lambda_* h > (1 + 2n)\pi/2$, where $\lambda_*$ is defined in (29). Further, $\lambda h$ takes all values from zero to $(1 + 2n)\pi/2$. Thus, Equation (26) is defined for $\lambda h \in (0, (1 + 2n)\pi/2)$, and the right-hand side of (26) is continuous in this interval. This means that a graph of the function in the right-hand side of (26) has at least $n$ intersections with tg $\lambda h$ in the left-hand side of (26) in interval $\lambda \in \left(\frac{\pi}{2h}, \frac{(1+2n)\pi}{2h}\right)$. $\quad\square$

Taking into account Statements 1–3 and the equivalence between Equation (21) and problem $\mathcal{P}$, one obtains the following result.

**Statement 4.** *If conditions* (27) *and* (30) *are fulfilled, then problem* $\mathcal{P}$ *has at least n eigenvalues* $\widehat{\gamma}_k \in (\gamma_*, \gamma^*)$, $k = \overline{1, n}$, *where*

$$\gamma_* = \sqrt{\frac{\varepsilon_2^2 - \varepsilon_1 \varepsilon_3}{2\varepsilon_2 - \varepsilon_1 - \varepsilon_3}}, \quad \gamma^* = \sqrt{\varepsilon_2 + \lambda_*'^2};$$

*here,* $\lambda_*'$ *is defined in* (24).

In fact, Statement 3 gives a sufficient condition for the existence of eigenmodes supported by the waveguide under consideration.

*3.5. Numerical Results*

In the calculations below, we used the following parameters: $\varepsilon_1 = \varepsilon_3 = 1$, $\varepsilon_2 = 11.7$, $\sigma^{(1)} = i \cdot 4.67 \cdot 10^{-4}$ S, $\sigma^{(3)} = -i \cdot 2.15 \cdot 10^{-17}$ S $\cdot$ m$^2 \cdot$ V$^{-2}$. The value of amplitude $A$ of the electric field is given in figures' captions.

It is worth giving a comment about the chosen values of the parameters. The linear part of the electric conductivity of graphene was calculated by virtue of the formula given in [22]. In this formula, there are three optional parameters: the wave frequency $\omega$, the chemical potential $\mu_c$, and the absolute temperature $T$; for calculating $\sigma_1$, we used $\omega = 2\pi \cdot 4 \cdot 10^{12}$ Hz, $\mu_c = 0.1$ eV, and $T = 300$ K; we stress that we neglected the real part of $\sigma_1$. Quantity $\sigma^{(3)}$ was calculated by virtue of the formula presented in [7] using the same parameters as for calculating $\sigma_1$.

In Figure 2, the dispersion curves of problems $\mathcal{P}$ (blue curves) and $\mathcal{P}_0$ (red curves) are presented. The dispersion curves were plotted as the dependence of a wave number (a propagation constant) on either the wave frequency $\omega$ or thickness $h$ of the waveguide. Since the statement of the problem does not involve $\omega$ explicitly due to the normalization by $k_0^2 = \omega^2 \mu_0 \varepsilon_0$, we plotted the dispersion curves as $\gamma$ vs. $h$.

The vertical line $h = 3$ in Figure 2 corresponds to the waveguide of thickness $3k_0^{-1}$. The intersection points of the dispersion curves with this line denoted by diamonds are the eigenvalues of the corresponding problems, and these eigenvalues, in turn, correspond to the propagation constants $\gamma k_0$ of the waveguide in the problems of electromagnetic wave propagation.

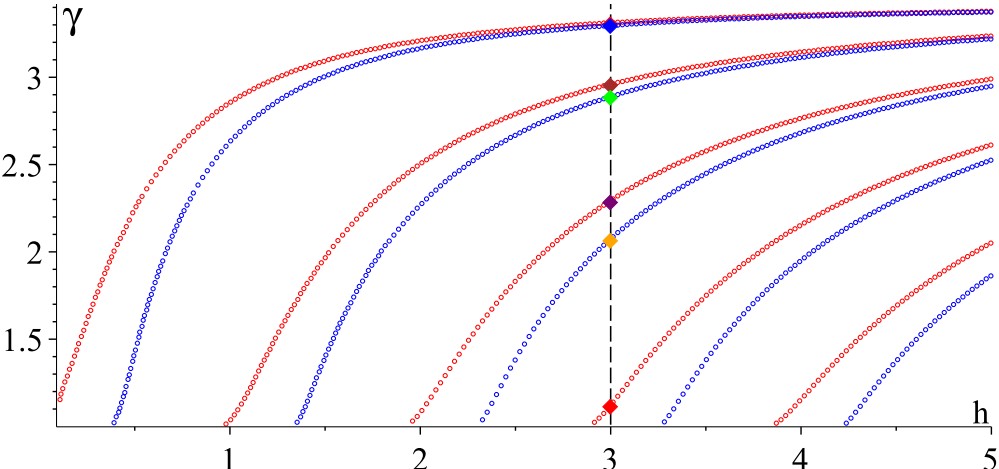

**Figure 2.** The first five dispersion curves of problems $\mathcal{P}$ (blue curves) and $\mathcal{P}_0$ (red curves); the value of the electric field at the boundary $x = 0$ is $A = 10^8$ V $\cdot$ m$^{-1}$. The diamonds denote the eigenvalues $\widehat{\gamma}_1 \approx 3.29$ (blue), $\widehat{\gamma}_2 \approx 2.88$ (green), and $\widehat{\gamma}_3 \approx 2.07$ (orange) of problem $\mathcal{P}$ and the eigenvalues $\widetilde{\gamma}_1 \approx 3.3$ (red), $\widetilde{\gamma}_2 \approx 2.95$ (brown), $\widetilde{\gamma}_3 \approx 2.29$ (purple), and $\widetilde{\gamma}_4 \approx 1.11$ (red) of problem $\mathcal{P}_0$.

In Figure 3, we plotted the eigenfunction $u(x; \widehat{\gamma}_3)$ of problem $\mathcal{P}$ and the eigenfunction $v(x; \widetilde{\gamma}_3)$ of problem $\mathcal{P}_0$. The eigenvalue $\widehat{\gamma}_3$ of problem $\mathcal{P}$ is a perturbation of eigenvalue $\widetilde{\gamma}_3$ of problem $\mathcal{P}_0$, and it can be shown that $\lim_{\sigma_3 \to +0} \widehat{\gamma}_3 = \widetilde{\gamma}_3$. Due to the closeness

of the eigenvalues $\widehat{\gamma}_3$ and $\widetilde{\gamma}_3$, it is natural to expect the closeness of the corresponding eigenfunctions $u(x; \widehat{\gamma}_3)$ and $v(x; \widetilde{\gamma}_3)$; Figure 3 demonstrates this clearly; in addition, it can be shown that $\lim_{\sigma_3 \to +0} u(x; \widehat{\gamma}_3) = v(x; \widetilde{\gamma}_3)$. In Figure 3, one can see that in the nonlinear case, the absolute value of the eigenfunction (tangential component of the electric field) at the boundary $x = h$ was significantly smaller than in the linear case. This means that the nonlinearity arising in graphene led to a greater localization of the electromagnetic field inside the waveguide. At the same time, the maximum and minimum values of the eigenfunction (tangential component of the electric field) in the nonlinear case were smaller in absolute value than in the linear case, and the extremum points shifted to the left relative to their positions in the linear case.

In Figure 4, we plotted the eigenfunction $u(x; \widehat{\gamma}_2)$ of problem $\mathcal{P}$ and the eigenfunction $v(x; \widetilde{\gamma}_2)$ of problem $\mathcal{P}_0$. The eigenvalue $\widehat{\gamma}_2$ of problem $\mathcal{P}$ is a perturbation of eigenvalue $\widetilde{\gamma}_2$ of problem $\mathcal{P}_0$, and it can be shown that $\lim_{\sigma_3 \to +0} \widehat{\gamma}_2 = \widetilde{\gamma}_2$. Figure 4 shows that the eigenmode corresponding to the nonlinear case was more localized than its linear counterpart. In addition, the maximum and minimum values of the nonlinear eigenmode were smaller in absolute value than the maximum and minimum values of the linear eigenmode, and the extremum points shifted to the left relative to their positions in the linear case.

It also seems interesting to learn at which conditions the discussed nonlinear effect in graphene became significant. In Figure 5, as well as in Figure 2, we plotted the dispersion curves of problems $\mathcal{P}$ (blue curves) and $\mathcal{P}_0$ (red curves); however, in the calculations, we used a smaller value for the amplitude of the electric field, namely $A = 2 \cdot 10^7$ V $\cdot$ m$^{-1}$. In this case, the dispersion curves of the (nonlinear) problem $\mathcal{P}$ were no longer strongly different from the dispersion curves of (linear) problem $\mathcal{P}_0$. In Figures 6 and 7, the eigenfunctions of the problems $\mathcal{P}$ and $\mathcal{P}_0$, corresponding to the eigenvalues denoted in Figure 5, are presented. It can be seen that the eigenmode corresponding to the nonlinear case was more localized within the waveguide than its linear counterpart; however, this effect was much weaker than the one demonstrated in Figures 3 and 4, and the reason is that the amplitude of the electric field was smaller. It is worth noting that the strength of the nonlinearity in graphene depends on the value of the nonlinearity coefficient $\sigma_3$ in the formula as well as on the amplitude of the incident wave; in accordance with the formula given in [7], the coefficient $\sigma_3$ was proportional to $\omega^{-3}$. Thus, in order to make the considered nonlinear effect more significant, one can either increase the amplitude of the incident wave or decrease its frequency.

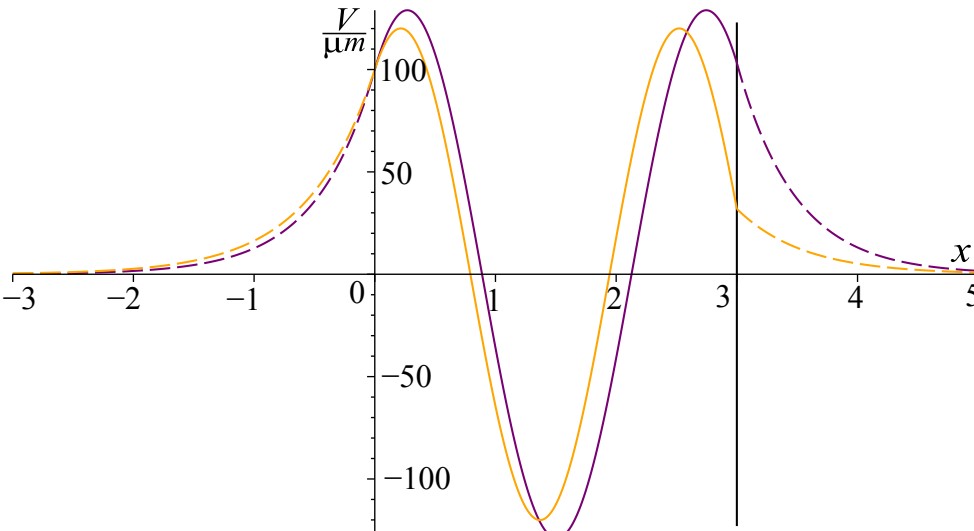

**Figure 3.** The orange curve presents the eigenfunction $u(x; \widehat{\gamma}_3)$ of problem $\mathcal{P}$, corresponding to the eigenvalue $\widehat{\gamma}_3 \approx 2.07$ denoted in Figure 2 by the orange diamond. The purple curve presents the eigenfunction $v(x; \widetilde{\gamma}_3)$ of problem $\mathcal{P}_0$, corresponding to the eigenvalue $\widetilde{\gamma}_3 \approx 2.29$ denoted in Figure 2 by the purple diamond.

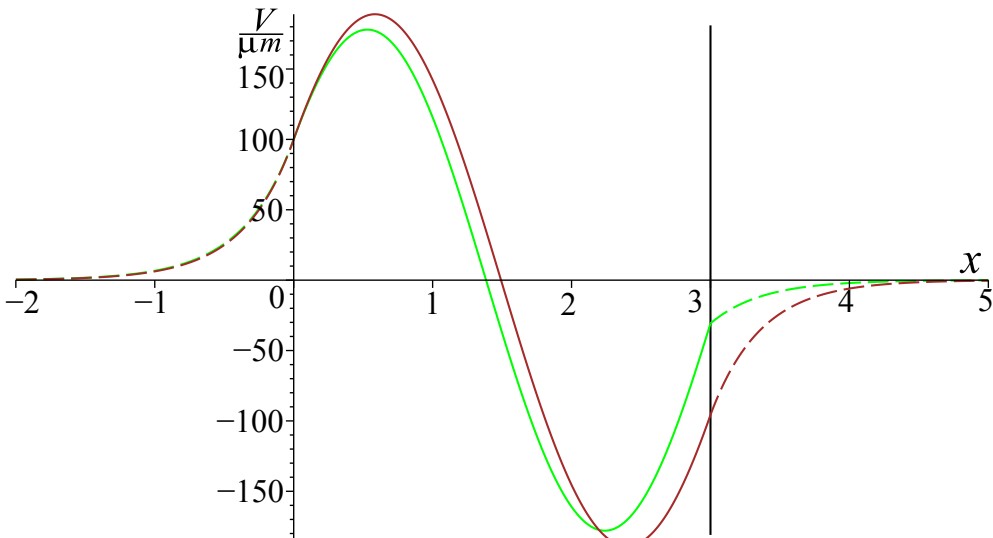

**Figure 4.** The green curve presents the eigenfunction $u(x; \widehat{\gamma}_2)$ of problem $\mathcal{P}$ corresponding to the eigenvalue $\widehat{\gamma}_2 \approx 2.88$ denoted in Figure 2 by the green diamond. The brown curve presents the eigenfunction $v(x; \widetilde{\gamma}_2)$ of problem $\mathcal{P}_0$ corresponding to the eigenvalue $\widetilde{\gamma}_2 \approx 2.95$, denoted in Figure 2 by the brown diamond.

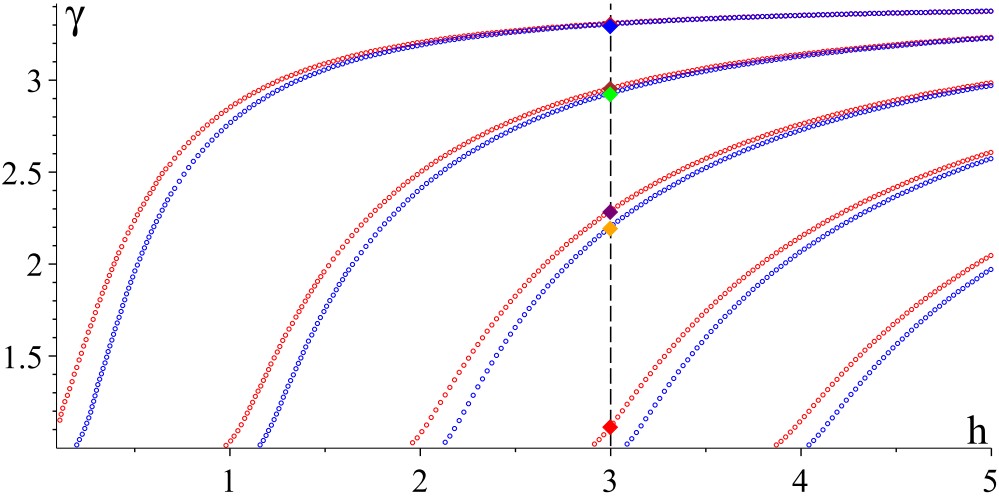

**Figure 5.** The first five dispersion curves of problems $\mathcal{P}$ (blue curves) and $\mathcal{P}_0$ (red curves); the value of the tangential component of the electric field at the boundary $x = 0$ is $A = 2 \cdot 10^7 \text{ V} \cdot \text{m}^{-1}$. The diamonds denote the eigenvalues $\widehat{\gamma}_1 \approx 3.3$ (blue), $\widehat{\gamma}_2 \approx 2.95$ (green), and $\widehat{\gamma}_3 \approx 2.19$ (orange) of problem $\mathcal{P}$ and the eigenvalues $\widetilde{\gamma}_1 \approx 3.3$ (red), $\widetilde{\gamma}_2 \approx 2.95$ (brown), $\widetilde{\gamma}_3 \approx 2.29$ (purple), and $\widetilde{\gamma}_4 \approx 1.11$ (red) of problem $\mathcal{P}_0$.

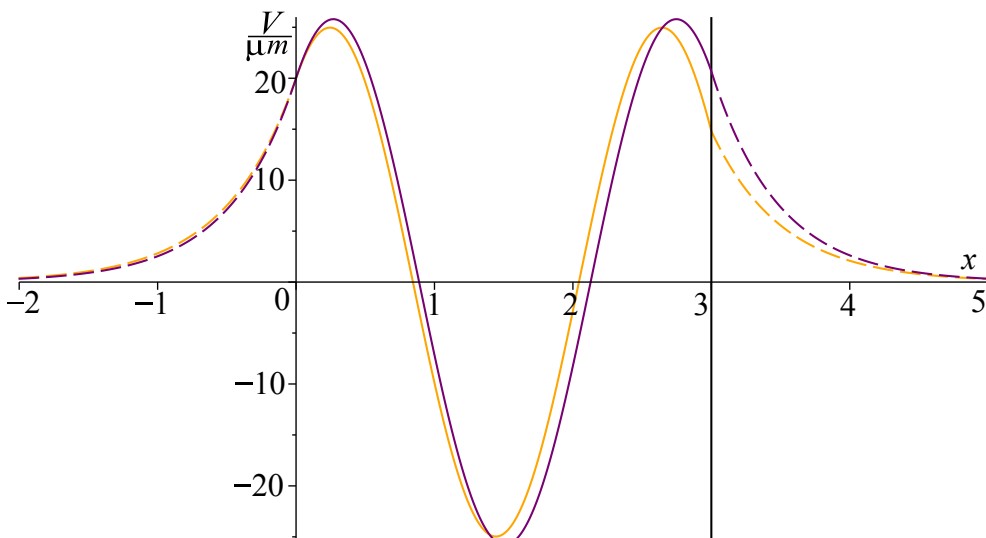

**Figure 6.** The orange curve presents the eigenfunction $u(x; \widehat{\gamma}_3)$ of problem $\mathcal{P}$, corresponding to the eigenvalue $\widehat{\gamma}_3 \approx 2.19$ denoted in Figure 5 by the orange diamond. The purple curve presents the eigenfunction $v(x; \widetilde{\gamma}_3)$ of problem $\mathcal{P}_0$, corresponding to the eigenvalue $\widetilde{\gamma}_3 \approx 2.29$ denoted in Figure 5 by the purple diamond.

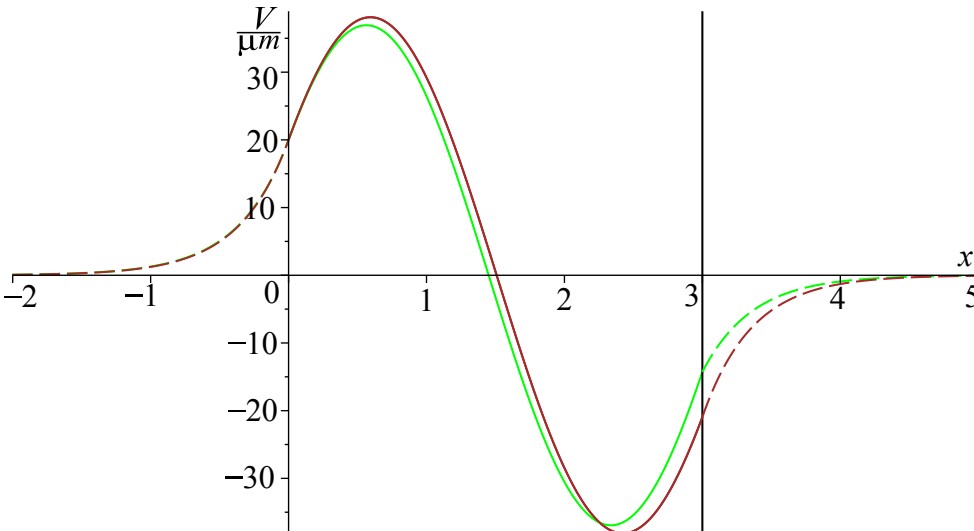

**Figure 7.** The green curve presents the eigenfunction $u(x; \widehat{\gamma}_2)$ of problem $\mathcal{P}$ corresponding to the eigenvalue $\widehat{\gamma}_2 \approx 2.92$ denoted in Figure 5 by the green diamond. The brown curve presents the eigenfunction $v(x; \widetilde{\gamma}_2)$ of problem $\mathcal{P}_0$ corresponding to the eigenvalue $\widetilde{\gamma}_2 \approx 2.95$, denoted in Figure 5 by the brown diamond.

## 4. Discussion

In the paper, the problem of the propagation of a monochromatic TE-polarized electromagnetic wave in a plane dielectric waveguide, covered with a layer of graphene on one side, is studied. The novelty of this study is that we take into account the nonlinearity of graphene, which becomes significant if the electromagnetic radiation is strong enough. The main result of this paper is the dispersion equation, which allows one to find, for a waveguide of a given thickness, a set of its propagation constants that determine the configuration of the so-called eigenmodes of the waveguide. We stress that whereas in most papers studying the problems similar to the one considered in this paper, the preference is given to numerical simulations, in this study we focused rather on obtaining qualitative results related to the solvability of the problem. Studying the dispersion equation, we

found sufficient conditions for the existence of several propagation constants (and their corresponding eigenwaves) of the considered waveguiding structure, see Statements 2 and 3. In other words, we proved the ability to propagate the TE waves localized in the plane dielectric waveguide covered with a graphene layer, taking into account the third-order nonlinearity of graphene affecting the wave propagation.

The numerical results, see Section 3.5, clearly demonstrate that the nonlinear effects in graphene have a significant affect on the electromagnetic wave propagation process, if the radiation is strong enough. For example, Figures 3 and 4 allow one to compare the component of the electric field in the linear (for $\sigma_3 = 0$) and nonlinear cases: one can see that in the nonlinear case, it was more localized within the waveguide, and its extremum points were shifted to the left relative to their position in the linear case.

In this section, we also want to briefly clarify why we were forced to neglect losses in our study. Let us look at the dispersion Equation (20). It is easy to notice that if $\sigma_1$ and $\sigma_3$ are complex numbers with a nonzero real part, then Equation (20) does not have real solutions $\gamma$, and we need to assume parameter $\gamma$ to be a complex number with a nonzero imaginary part. However, that leads to the following problem. In Formulas (2), the functions $E_y$, $H_x$, and $H_z$ depend only on one spatial variable: the transverse coordinate $x$. This choice of the fields is possible only in the case of real $\gamma$. Indeed, substituting the fields **E** and **H** with components (2) into Formula (4), with $\sigma_g$ defined in (5), we obtain the equation that depends on $x$ and does not depend on $z$. To be more precise, the multiplier $e^{i\gamma z}$ is reduced on both sides of the equations, and the term $|\mathbf{E}|$ does not depend on $z$ as $|e^{i\gamma z}| = 1$ for a real $\gamma$. At the same time, if $\gamma$ is not real, then $|e^{i\gamma z}|$ is a function with respect to $z$. This means that components $E_y$, $H_x$, and $H_z$ depend not only on $x$ but also on $z$. This contradicts the choice of these components as functions depending only on $x$. All this, however, does not mean that it is not possible to consider a complex $\gamma$ in a waveguiding problem with a nonlinear graphene layer. We just say that for a complex $\gamma$, the problem cannot be solved correctly mathematically for fields (1) and (2). To overcome the described problem, we need $\sigma_1$ and $\sigma_3$ to be purely imaginary, since in this case, the dispersion Equation (20) might have real solutions.

In this paper, we focus only on TE-waves and do not study the ability to propagate TM-polarized waves in the considered waveguiding structure. However, this does not mean that the TM case is less important and less interesting than the TE one. The reason why we do not study TM waves here is that the governing equations as well as the boundary conditions for the TE and TM-waves are different, and the analytical approach, developed in this paper to deal with the TE-waves, needs to be strongly modified for TM-polarized waves. So, we strongly believe that it is more convenient to study these two types of waves separately.

An essential continuation of this research will be the study of waveguiding structures of a more complex configuration. For example, it is interesting to study the problem of the propagation of a monochromatic TE-polarized wave in a plane waveguide covered on both sides by layers of graphene and to take into account the arising nonlinear effects at both boundaries of the waveguide. We stress that such a problem is very important from a practical point of view, as well as rich in mathematical content, because it involves two boundary conditions, which are nonlinear with respect to the sought-for function; in other words, there arises a new class of boundary value problems with nonclassical (nonlinear) boundary conditions. Finally, we stress that circular cylindrical waveguides combined with graphene are of great importance.

**Author Contributions:** Methodology, Y.S.; software, S.T.; investigation, S.T. and Y.S.; writing—original draft preparation, S.T.; writing—review and editing, Y.S.; visualization, S.T.; supervision, Y.S. All authors have read and agreed to the published version of the manuscript.

**Funding:** The research was funded by the Russian Science Foundation under the project 20-11-20087.

**Institutional Review Board Statement:** Not applicable.

**Informed Consent Statement:** Not applicable.

**Data Availability Statement:** Not applicable.

**Conflicts of Interest:** The authors declare no conflict of interest.

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
