# Peer review of "The Nonlinear Eigenvalue Problem of Electromagnetic Wave Propagation in a Dielectric Layer Covered with Graphene"

_photonics, doi:10.3390/photonics10050523_

Round 1

Reviewer 1 Report

This mathematical treatment of the eigenvalue problem of 1D waveguide containing one non-linear graphene layer as a boundary deserves be documented in literature. The formulism could be useful when designing graphene waveguides which operates with strong light field. But in this manuscript, the writing style could concentrate more on the physics of modes in the waveguide. My comments are listed below.

1. The major drawback of this formulism is that the eigenmodes are assumed to be lossless but in fact they are not. It is better to state it candidly in the Abstract.

2. I believe the assumption of epsilon1 <= epsilon3 is unnecessary.

3. Since nonlinear effect is an important concern, Figures showing how the eigenmodes change with the light intensity would more directly demonstrate the importance of this work.

4. Section 2, which contains only three sentences, could be merged in Section 3.

Reviewer 2 Report

See report.

Author Response

Thank you for your report. We redesigned and extended section Discussion which in particular summarizes the obtained results and contains the main conclusions of the study. Besides this, we made some improvements on the Abstract and the Introduction. We believe that now the results and conclusions of this study are presented more accurately.

Reviewer 3 Report

This manuscript entitled “Nonlinear eigenvalue problem of electromagnetic wave propagation in a dielectric layer covered with grapheme (Manuscript ID: photonics-2350852)” by Y. Smirnov et al., focuses on the problem of a monochromatic TE-polarized wave propagation in a plane dielectric layer filled with a homogeneous isotropic medium, takes into account the significant nonlinearity resulting from the interaction of the electromagnetic wave with the charge carriers in the graphene layer. The paper is well organized, we would recommend the acceptance of this manuscript for publication in Photonics with minor revisions if the following concerns can be addressed.

1. It is recommended that the authors compare the results of the manuscript with those of TM-polarized waves.

2. The letters and numbers in Figures 3 and 4 need to be enlarged.

3. Line 53 on Page 2: The “section” is suggested to be changed as “Section”;

Line 68 on Page 2: The “fig. 1” is suggested to be changed as “Fig. 1”; and so on.

Please check them,

and some other common problems that need to be addressed by the authors.

4. Line 190 on Page 7: What is â–¡?

5. The format of the references needs to be further improved.

For example,

In reference [8], the “Applied Physics Letters” should be changed to “Appl. Phys. Lett.”,

Please check them,

and some other common problems that need to be addressed by the authors. 

Minor editing of English language required.

Reviewer 4 Report

In this article, the authors focus on the propagation of monochromatic TE polarized waves in a planar dielectric layer filled with uniformly isotropic media. In the model, one of the boundaries of the waveguide is covered by a graphene layer. The studied model takes into account the significant nonlinearity resulting from the interaction of the electromagnetic wave with the charge carriers in the graphene layer. I have some comments to do after reviewing this manuscript, details are as follows:

1.      In order to better highlight the advantages of this work, the author needs to provide a table to compare related work.

2.      The abstract of this paper is too simple and needs further supplementation.

3.      The second part of this paper, Materials and Methods, was written too simply by the author and did not provide a detailed introduction to the relevant methods and physical models

4.      The author did not provide corresponding numbers for some formulas in the article.

5.      Some image authors did not provide a detailed explanation, such as Figure 4

6.      The introduction can be improved. The articles related to some applications of graphene materials should be added such as Sensors 2022, 22, 6483; ACS Sustain. Chem. Eng. 2015, 3, 1677–1685; Diamond & Related Materials 128 (2022) 109273; Results in Physics 48, 2023, 106420.

7.      Please check the grammar and spelling mistakes of the whole manuscript.

Please check the grammar and spelling mistakes of the whole manuscript.

Reviewer 5 Report

In this paper, the authors report on analytic calculations they performed to get the dispersion curve of a plane dielectric waveguide covered with a graphene layer at one side. The main showcased result concerns the non-linear response of the graphene in terms of its surface conductivity, which relates to the square of the electric field norm. The authors obtain the dispersion curve including the non-linearity aforementioned after formulating and solving the corresponding boundary eigen-value problem. It is shown indeed the effect of non-linearity in terms for instance of eigenmodes becoming more localized, which consists the sole interesting feature of the results.

1)     I find the introduction to be insufficient, and I struggle to understand what is really original in the proposed work? The authors should include more recent literature, and state clearly their added value.

2)     The authors work essentially with TE polarization, why is this? Why not consider TM polarization as well?

3)     It is assumed by the authors that losses can be negligeable in graphene, what could possibly justify this? The graphene conductivity sigma is complex, so unless the authors assume working in some frequency range where the imaginary part of sigma is negligeable, I am not convinced that the assumption is correct.      

The english writing level  is OK overall, but it is best if the authors put some additional efforts to improve the manuscript writing quality. 

Round 2

Reviewer 3 Report

None

Reviewer 4 Report

 Accept in present form.

Reviewer 5 Report

The authors followed recommendations and answered all my points, so the article can be published in its form.

The Ensglish is OK, though some moderate or minor editing from the editorial bord might be necessary to meet the journal writing style and standards.